# Does the COVID Pandemic Modify the Antibiotic Resistance of Uropathogens in Female Patients? A New Storm?

**DOI:** 10.3390/antibiotics11030376

**Published:** 2022-03-10

**Authors:** Cristian Mareș, Răzvan-Cosmin Petca, Aida Petca, Răzvan-Ionuț Popescu, Viorel Jinga

**Affiliations:** 1Department of Urology, “Carol Davila” University of Medicine and Pharmacy, 050474 Bucharest, Romania; cristian.mares@drd.umfcd.ro (C.M.); razvan-ionut.popescu@drd.umfcd.ro (R.-I.P.); viorel.jinga@umfcd.ro (V.J.); 2Department of Urology, “Prof. Dr. Th. Burghele” Clinical Hospital, 050659 Bucharest, Romania; 3Department of Obstetrics and Gynecology, “Carol Davila” University of Medicine and Pharmacy, 050474 Bucharest, Romania; 4Department of Obstetrics and Gynecology, Elias University Hospital, 011461 Bucharest, Romania

**Keywords:** UTIs, females, AMR, uropathogens, COVID, pandemic, resistance, *Escherichia coli*, *Klebsiella*

## Abstract

Urinary tract infections (UTIs) represent a common pathology among female patients, leading to overprescribing antibiotics, globally. The emergence of the COVID-19 pandemic has dramatically increased the incidence of this particular viral pneumonia with secondary bacterial superinfection, resulting in continuous therapeutic or prophylactic recommendations of antibiotic treatment; thus, an updated analysis of current antimicrobial resistance among uropathogens is mandatory. This cross-sectional retrospective study conducted in two university hospitals in Bucharest, Romania analyzed 2469 positive urine cultures, among two different periods of 6 months, before and during the COVID-19 pandemic. The most common pathogen was *Escherichia coli* 1505 (60.95%), followed by *Klebsiella* spp. 426 (17.25%). *Enterococcus* spp. was the leading Gram-positive pathogen 285 (11.54%). In gram negative bacteria, in almost all cases, an increased in resistance was observed, but the highest increase was represented by quinolones in *Klebsiella* spp., from 16.87% to 35.51% and *Pseudomonas* from 30.3% to 77.41%; a significant increase in resistance was also observed for carbapenems. Surprisingly, a decrease in resistance to Penicillin was observed in *Enterococcus* spp., but the overall tendency of increased resistance is also maintained for gram positive pathogens. The lack of data on the influence of the COVID-19 pandemic on uropathogens’ resistance promotes these findings as important for every clinician treating UTIs and for every specialist in the medical field in promoting reasonable recommendations of antibiotic therapies.

## 1. Introduction

Urinary tract infections (UTIs) represent a common bacterial disruption that affects the entire urinary tract in males and females, from the distal bladder and urethra to the kidney via ureters. It is considered one of the most frequent causes of infectious presentation to medical service, affecting over 150 million patients yearly [1]. UTIs represent more than 1% of the total number of ambulatory visits; moreover, in emergency medicine, it is estimated that there are more than 3 million visits every year for symptoms of a UTI [2]. In the general population, the prevalence of these infections increases linearly with every decade, except for the spike of the age group 14–24 years old in young women [3]. Considering UTIs an essential issue in continuously aging populations, studies suggest that women over 65 years old are more likely to experience an episode of UTI, accounting for approximately 20% of all cases. In contrast, only half of the general population is at risk [4]. Overall, more than 40% of all women will be diagnosed with a UTI during their lifetime, while one in three females will be diagnosed before turning 24-years-old [5]. The tremendous amount of new and recurrent UTI cases burdens the economic healthcare system, with more than 1.6 billion dollars in annual costs for diagnosing and treatment in the United States alone [6].

Treatment of UTIs commonly involves antibiotics to combat the infection, resulting in long-term alteration in sensibility and resistance patterns of various bacterial strains implicated in the etiology. The ubiquitous most common bacteria involved in these particular infections is *Escherichia coli* [7]. Various European and local studies suggest other Gram-negative microorganisms responsible for UTIs, such as *Klebsiella* spp., *Proteus* spp., or *Pseudomonas* spp. [8,9,10]. Gram-positive bacteria are also involved, such as *Enterococcus* spp., which can occur in almost 20% of the cases [11]. At the same time, *Staphylococcus* spp. is more frequent in elderly and institutionalized patients [12], provoking symptoms of infection similar to *E. coli* but usually more severe [13].

EAU’s latest guidelines recommend adjusting antibiotic therapy to local resistance trends [14]. Unfortunately, data determining the evolutive character of these patterns on Romanian females on a prolonged period are not consistent. Recent evidence shows *E. coli* as the most frequent uropathogen with a leading resistance to Levofloxacin and β-lactams. *Klebsiella* spp. follows with high resistance to aminopenicillins, Nitrofurantoin, and cephalosporins [15]. Both strains still demonstrated preserved sensitivity to carbapenems and aminoglycosides. *Enterococcus* spp. was the leading Gram-positive bacteria proving significant resistance to fluoroquinolones and β-lactams, with preserved sensitivity to vancomycin, fosfomycin, and nitrofurantoin [15].

Considering the 2019 epidemic outbreak caused by COVID-19, a viral infection, the treatment in a substantial number of patients implied antibiotics use. Studies suggest that over 70% of the patients diagnosed with this disease and admitted to hospitals received at least one antimicrobial agent, prophylactically, despite less than 10% presenting a bacterial infection [16,17]. Moreover, the alarming information from media and the internet that invades everyday life has raised levels of antibiotic consumption despite the lack of recommendation, accounting for the fact that more than 65% of the patients presenting to hospitals with symptoms of the viral infection have already resorted to taking antibiotics [18]. This tendency in the overconsumption of antibiotics combined with already published data on antibiotic resistance and multi-drug resistance [19,20] directly impacts the patient’s outcome, leading to higher morbidity and mortality rates. Studies suggest that, following this trend, by 2050, over 10 million deaths are expected to occur because of this [21].

Data on antibiotic resistance from the pre-pandemic period and throughout the COVID-19 outbreak are relevant to precisely determine the evolution of this critical epidemiological issue, especially on uropathogens due to their high versatility in resistance profile, representing key factors in initiating the study.

## 2. Results

A total number of 2469 bacterial strains met the inclusion criteria to be admitted in the study, summing more than 10^5^ CFU/mL. The two hospitals involved contributed variably, as follows: “Prof. Dr. Th. Burghele” Clinical Hospital (BCH) 1184 patients (47.95%) and Elias University Hospital (EUH) 1285 patients (52.04%). In the pre-pandemic time, the total number of recorded cases was 1505 (60.95%), while, during the pandemic time, 964 cases were registered (39.04%) as follows: pre-pandemic, at BCH, 739 cases (29.93%) were recorded, while, at EUH, 766 cases (31.02%) were recorded; during the pandemic period, at BCH, 445 cases (18.02%) were recorded, while, at EUH, 519 cases were reported, representing (21.02%). Gram-negative bacteria represented 2132 (86.35%) from the total number of strains, while Gram-positive bacteria represented only 337 cases (13.64%). *Escherichia coli* was identified as the most common Gram-negative microorganism, representing 1505 (60.95%), followed by *Klebsiella* spp. with 426 strains (17.25%), *Proteus* spp. with 137 strains (5.54%) and *Pseudomonas* spp. with 64 strains (2.59%). *Enterococcus* spp. was determined to be the most common Gram-positive bacteria summing 285 cases (11.54%), followed by *Staphylococcus* spp. with 52 strains (2.1%). Extensive data reported on Gram characteristics of all strains enrolled in the study for each hospital center is represented in Table 1.

UTIs in women are characterized by a variable inconsistency in terms of age stratification due to a multitude of factors that influence its’ incidence, such as sexual activity and hormonal status. In the overall studied population, there is a linear rise in incidence that is directly proportionate with age in almost all strains. In women under 40 years old, considered the most sexually active, the incidence in the pre-pandemic period was 184 (12.22%) and, during pandemic 120 (12.44%); they were followed by the middle-aged women in 40-55 years old group, representing, in the pre-pandemic time, 186 patients (12.35%) and, during the pandemic period, 172 patients (17.84%). The most frequent cases of UTIs were observed in post-menopausal women >55 years old, representing, in the pre-pandemic time, 1135 (75.41%) patients and, during the pandemic, 673 patients (69.70%). The exception of this trend was observed for *Escherichia coli*, the most frequent uropathogen in both groups, in the pre-pandemic time; the highest incidence was observed in mature and elderly women >55 years old, representing 705 patients (46.84%), followed by young women <40 years old, representing 116 patients (7.70%); the lowest incidence was observed in middle-aged women, representing 104 patients (6.91%). Extensive data on the variation of UTIs in female patients is represented in Table 2.

*Escherichia coli* was the most frequent uropathogen in both studied groups and in both centers; the highest overall resistance combined with a significant growth during COVID pandemic was observed for Amoxicillin-Clavulanic Ac., from 14.27% in the pre-pandemic time to 21.37% during the pandemic, followed by Levofloxacin from 27.45% to 28.79%, Ceftazidime from 7.13% to 8.1% and Nitrofurantoin from 4.86% to 6.72%. Considering the antibiotics with the most preserved sensitivity, Fosfomycin led the selection, from 91.56% pre-pandemic to 91.89% during the pandemic, followed by Amikacin, from 87.45% to 81.03%, Ceftazidime, from 82.81% to 77.41% and Levofloxacin, from 63.45% to 56.72%. The only drug-classes of antibiotics that presented a higher sensitivity in the latest group compared with the pre-pandemic period were Carbapenems, Fosfomycin and Nitrofurantoin. *Klebsiella* spp. was the second highest Gram-negative uropathogen in both groups, with the highest resistance to Amoxicillin-Clavulanic Ac. from 29.21% to 38.79%, followed by Ceftazidime from 16.04% to 25.13%, Levofloxacin from 16.87% to 35.51% and Nitrofurantoin from 15.63% to 20.76%. In all cases, a drop in the sensitivity rates was observed for each antimicrobial for this pathogen in both period groups. Detailed statistics for these two Gram-negative uropathogens are represented in Table 3.

*Pseudomonas* spp. presented the most dramatic falls in the sensitivity patterns; the highest resistance growth was observed for carbapenems: imipenem from 18.18% to 67.74% and Meropenem from 18.18% to 64.51%. Furthermore, a significant rise in resistance was observed for Levofloxacin from 30.03% to 77.41%, Ceftazidime from 27.27% to 67.74%, Amikacin from 15.15% to 51.61%, and aztreonam from 9.09% to 41.93%. A drop in the sensitivity rates was observed for this pathogen in all antimicrobials tested. The most common urease-producing bacteria, *Proteus* spp., presented higher resistance and significant growth to amoxicillin-clavulanic ac. from 27.5% to 28.07%, followed by Levofloxacin from 23.75% to 33.33%, Ceftazidime from 15.0% to 15.78% and Amikacin from 8.75% to 12.28%. Except for Meropenem, which presented growth in the sensitivity from 71.25% to 73.68%, it faces a drop in this matter for all other antibiotics. A more detailed representation of resistance and sensitivity patterns for these two Gram-negative uropathogens is displayed in Table 4.

The most common Gram-positive uropathogen, *Enterococcus* spp., shows a rise in the resistance profiles for all the antibiotics it was tested for, except for, surprisingly, Penicillin, the resistance rates of which dropped from 29.31% to 25.53%; the highest resistance was to Levofloxacin, from 31.93% to 35.1%. The highest sensitivity rates were observed for vancomycin, from 89.52% to 84.04%, followed by Nitrofurantoin, from 86.91% to 91.48%, linezolid from 86.38% to 81.92%, and ampicillin from 76.96% to 79.78%. *Staphylococcus* spp. presented the highest resistance to Penicillin, which varies from 45.45% in the pre-pandemic time to 47.36% during the pandemic, followed by Levofloxacin from 21.21% to 26.31%; the third most resisted antibiotic for this pathogen in the pre-pandemic time, trimethoprim/sulfamethoxazole, presented a drop in the resistance rates from 18.18% to 5.26%, with an increased sensitivity from 57.57% to 89.47%. Moreover, other antibiotics that showed an increase in antibiotic susceptibility are Nitrofurantoin, from 72.72% to 78.94% and linezolid, from 75.75% to 89.47%. Detailed statistics on Gram-positive bacteria are presented in Table 5.

## 3. Discussion

Uropathogens determine the urinary tract colonization, leading to inflammation and infection at various levels, which is a key factor of morbidity due to UTIs, which affect millions of people yearly, irrespective of gender and age. UTIs have a higher prevalence in women compared to men, considering an overall occurrence of more than 80% in females; recurrence is also more present in women; studies suggest that almost 30% of them will experience another episode after six months apart from the first episode, while 48% of repeated episodes will occur within one year [22]. The lower urinary tract anatomy implies a higher susceptibility of UTIs in females, considering the shorter distance between the perianal area, vaginal cavity, and urethral opening, and the shorter urethra compared to males; all the above constitute a favorable path for extrinsic bacterial inoculation.

General risk factors are associated with a higher prevalence of UTIs, such as female gender, sexual activity, diabetes, a prior episode of UTI, genetic susceptibility or obesity [23]. In females, the risk factor varies with age stratification, sexual behaviour and hormone status; in premenopausal women, various elements of susceptibility are stated, such as frequent sexual intercourse of four or more times per week, changes in bacterial flora, use of spermicides or diaphragm, history of UTIs in childhood or family history, changing to a new sexual partner within one year, lack of postcoital voiding or poor hygiene [24,25]. In postmenopausal women, studies suggest that urinary incontinence, lack of trophic hormones of the genitourinary tract, anterior vaginal wall prolapse, urinary catheterization, or increased postvoid residual volume are the main risk factors for the bacterial infection colonization of the urinary tract [26]. Multiple international surveillance programs are designed to evaluate the level of uropathogens’ resistance at precise times. Still, locally acquired data is requested to better comprehend the dynamics of resistance status. Nevertheless, observing the evolution of sensitivity and resistance at different points in time is necessary to determine whether the trends are favorable or extensive control of antimicrobial resistance (AMR) is still required.

Additionally, preliminary data suggest that the COVID-19 pandemic has influenced the resistance patterns of various bacteria due to overusing of self-administered over-the-counter antibiotics, as well as physician-prescribed drugs for inpatients admitted for the viral infection. Knowledge of the resistance trend of the most common uropathogens is of the utmost importance, allowing for the best standard of care in combating these infections. Thus, results from a descriptive “cross-sectional” retrospective study, designed in two university-teaching hospitals from the capital city of Romania, acquiring data regarding UTIs’ pathogens involvement and resistance patterns, in two distinct periods, pre- and during-pandemic, are provided.

### 3.1. Frequency of Bacterial Strains Implicated in the Etiology of UTIs

The uropathogens found in females’ urine samples are most often Gram-positive, accounting for between 85% to 88%, *Escherichia coli* outpacing the others by a large margin, at 60.95%. In fact, this enterobacteria leads the frequency of uropathogens in almost all studies, from all continents accounting for various percentages: Switzerland, Central Europe—74.6% [27], Al-Kharj, Saudi Arabia—70.4% [28], Seoul, South Korea—87.3% [29], USA, North America (A wationwide analysis)—72% [30], KwaZulu-Natal, South Africa—81.25% [31]. The second most common Gram-negative bacteria are *Klebsiella* spp., representing 17.25% of the total strains; similar findings were reported regionally, in Hungary [32], accounting for 13.4% of the total inpatients’ urine samples, comparable with other findings in the Middle East, such as Turkey—11.2% [33]. *Pseudomonas* spp. and *Proteus* spp. hold the 3rd and 4th places, respectively, in terms of frequency of uropathogens with interchangeable proportions; this study highlights *Proteus* spp. as more frequent, representing 5.54%, compared to *Pseudomonas* spp.—2.59%, but this is not applicable with other findings. Jan Hrbacek et al. recently published a paper analyzing urine samples over nine years, observing more strains of *P. aeruginosa*—7.3% than *Proteus* spp.—6.2% [34]. *Enterococcus* spp. had the higher incidence in Gram-positive bacteria, followed by *Staphylococcus* spp, the first representing 11.54%, while the latter only 2.1% of the total amount of strains. Various data are available in the literature, with fluctuating incidence from an overall of 8.2% of all Gram-positive cocci from Urmi et al. [35] to 21.3% only of *Enterococcus faecalis* from Shrestha et al. [36].

### 3.2. Evolution of Resistance Patterns of Gram-Negative Bacteria

Current data on antibiotic resistance trends between European countries [34,37,38] imply discouraging results regarding resistance rates accounting for *E. coli* for aminopenicillins, fluoroquinolones, and trimethoprim/sulfamethoxazole; promising sensibility profiles were observed for aminoglycosides, carbapenems and cephalosporins. *Klebsiella* spp was also a leading Gram-negative uropathogen, presenting high resistance to cephalosporins, fluoroquinolones, and Nitrofurantoin, with an overall sustained response to polymyxin B, colistin, and carbapenems. *Pseudomonas aeruginosa* showed relatively high susceptibility to piperacillin/tazobactam, aminoglycosides, and carbapenems, while *Proteus* spp. highlighted good response to aminopenicillins, piperacillin/tazobactam, third-generation cephalosporins, and Amikacin.

As more antibiotics are prescribed yearly, the expected evolution of antimicrobial resistance tends to rise. Thus, comprehensive research for each uropathogen was realized. *Escherichia coli* presented the highest resistance to Levofloxacin R = 27.45% in the pre-pandemic period and 28.79% during the pandemic. The highest raise was observed for Amoxicillin–Clavulanic Ac. from R = 14.27% to 21.37%; a significant increment was observed for Nitrofurantoin, a first-line drug, from R = 4.86% to 6.72%. In an extensive work from Hungary, Mario Gajdacs et al. followed the evolution of AMR over ten years [39]. They observed significant exacerbation of resistance in multiple classes of antibiotics, similar to our result, such as fluoroquinolones in outpatients from 2008 R = 13.28% to 2017 R = 25.95%, inpatients from 2008 R = 20.19% to 2017 R = 33.25%; aminoglycosides in outpatients from 2008 R = 3.17% to 2017 R = 13.10%, inpatients from 2008 R = 7.82% to 2017 R= 13.10%; cephalosporins in outpatients from 2008 R = 8.99% to 2017 R = 9.70%. Contrarily, in Nitrofurantoin’s case, the Hungarian authors observed a decrease in the resistance patterns, as they highlighted resistance in outpatients from 2008 R = 4.99% to 2017 R = 1.03 and inpatients from 2008 R = 7.98% to 2017 R = 1.39% [39]. In our study, none of the tested antibiotics presented lower resistance rates in *Escherichia coli* in the second period compared to the first one.

In terms of resistance, the situation for *Klebsiella* spp. is even worse. The highest resistance was observed for aminopenicillins—Amoxicillin-Clavulanic Ac. R = 38.89%, but the maximal raise in resistance was observed for fluoroquinolones—Levofloxacin from pre-pandemic R = 16.87% to 35.51%. A study designed in Manisa, Turkey [33] over a period of 5 years and published last year also observed the highest resistance to Ampicillin R = 76.02%, followed by Trimethoprim/Sulfamethoxazole R = 39.85% and Nitrofurantoin R = 37.61%. Contrarily, they noted meager resistance to Levofloxacin R = 3.15% and Meropenem R = 2.76%. This study observed resistance of more than five times higher for Meropenem than the Turkish study, varying from 7.4% in the pre-pandemic period to 11.47% during the pandemic.

Our calculation showed the highest resistance rates in *Pseudomonas* spp.’ case, with the highest rise of all uropathogens; Levofloxacin leads the resistance race, from R = 30.3 to R = 77.41%, followed by Ceftazidime, Imipenem, Meropenem, and Amikacin. During nine years from the Czech Republic, long-term research published last year [34] presented lower resistance rates in all the antibiotics tested for this Gram-negative strain, as follows: Ciprofloxacin R = 38.1%, Ceftazidime R = 18.7%, Imipenem R = 15.6%, Meropenem R = 31.8%, Amikacin R = 9.3%. The lower resistance in our study was observed for Nitrofurantoin R = 9.67%; thus, it still represents a viable option in treating this type of uncomplicated UTI.

The most important urea-producing bacteria involved in UTIs, *Proteus* spp., accounts for a little over 5% of the total strains, but it presents important morbidity. The highest resistance was observed for Levofloxacin R = 33.33%, followed by Amoxicillin-Clavulanic Ac. R = 28.07%, Ceftazidime R = 15.78% and Amikacin R = 12.28%. This year, similar results were reported in Brasil [40] in a paper that studied 92 articles published recently, involving more than 3385 positive urine samples from females: fluoroquinolones (Ciprofloxacin R = 21.56%, Amoxicillin-Clavulanic Ac. R = 25.49%), cephalosporins (Cefuroxime R = 19.69% and Ceftriaxone R = 18.30%). The lowest resistance in the South American paper was determined for Amikacin R = 5.88%, while this study highlighted carbapenems (Meropenem R = 1.75%).

### 3.3. Evolution of Resistance Patterns of Gram-Positive Bacteria

*Enterococcus* spp. was the most frequent Gram-positive uropathogen and the third overall; it presented the highest resistance to Levofloxacin R = 35.1%, followed by Penicillin R = 25.53% and Ampicillin R = 18.08%. The highest growth in resistance was observed for Fosfomycin from R = 0.52% to R = 6.38%. The only antibiotic that presented a lower resistance during the pandemic compared to the pre-pandemic time was Penicillin, from 29.31% to 25.53%. A recent paper published last year from Hungary that evaluated the resistance to antibiotics of Gram-positive bacteria over the previous ten years [41] presented similar results for fluoroquinolones R = 33% for inpatients between 2013–2017, but significantly lower values for the rest of antibiotics tested, as follows: Linezolid R = 0%, compared to R = 2.09 and R = 4.25% in our study; Ampicillin between R = 0.2% and R = 0.4%, compared to R = 17.08% and R = 18.08% in our study; Vancomycin between R = 0.1% and R = 0.3%, compared to R = 1.57% to R = 2.12%.

*Staphylococcus* spp. represent the rarest finding in urine samples of all uropathogens and Gram-positive strains. We discovered a drop in the resistance trends, such as Amikacin from R = 3.03% to R = 0%, Trimethoprim/Sulfamethoxazole from R = 18.18% to R = 5.26%, Linezolid from R = 6.06% to R = 0% and Nitrofurantoin from R = 6.06% to 5.26%. Contrary to our findings, the Hungarian paper highlighted a critical resistance level for Penicillin between R = 95.2% and R = 96.9% from 2013 to 2017; they also presented high rates of resistance for Trimethoprim/Sulfamethoxazole R = 27.1%, and the fluoroquinolones Ciprofloxacin R = 25.6% and Amikacin R = 14.2% [41]. Comparable to our results, they acknowledged no resistance to Linezolid and Vancomycin.

### 3.4. The Implication of COVID Pandemic in AMR of Uropathogens

The ongoing viral COVID-19 pandemic, which causes the severe acute respiratory syndrome coronavirus 2 (SARS-CoV 2), leads to an enormous strain on the economic and healthcare systems globally. This sanitary endeavor imposed several prevention measures, such as social distancing, sanitary mask-wearing, avoiding crowded areas, proper hand hygiene, and active identification and quarantine of close contacts. Therefore, these accurate measurements could have contributed to a decline in the spread of the virus and protected people from getting infected with other types of common infections, such as seasonal influenza, tuberculosis, or pneumococcal disease.

However, COVID-19 could represent an exacerbation factor for multiple reasons in terms of AMR. First, increased self-medication of over-the-counter antibiotics leads to overconsumption, presenting a higher risk of subtherapeutic doses and shortened courses of these drugs, implying negative trends of AMR and leading to increased mortality [42,43]. Secondly, patients that solicited medical help received different options of antibiotic therapy. Often, prescribing antibiotics was handy to exclude bacterial infectious diseases that mimic the symptoms of COVID infection such as pneumonia or as an attribute in fighting different complications that can occur during the viral episode of infection [42], even if multiple surveys suggest that co-infection rates are low [44,45]. Furthermore, the World Health Organisation (WHO) recommends not to provide antimicrobial therapy or prophylactic doses in patients with mild or moderate disease stages unless there is a clear clinical indication in that direction [46]. For example, an analysis of 154 studies aimed to estimate antibiotic prescription prevalence in patients infected with COVID, gathering data from over 30,000 patients, highlighted that three-quarters of them received antibiotics during the viral infection episode [47]. These results demonstrated that unnecessary antibiotic use was considerably high in these cases. Like the aforementioned study, similar recent papers indicate that the viral epidemic could amplify AMR in the near future [48,49,50,51,52]. A Belgian survey from this year involving 164 COVID positive patients, 15.2% of whom were admitted to the ICU and 15.9% of whom died during hospitalization, underlined that 61% of the total number also received an antibiotic treatment which, in the end, did not reduce the mortality rate [53]. Contrarily, some authors suggest that this significant public health problem is not at risk of exacerbation. It could lead to further positive steps due to an improvement in infection prevention and control practices, and to restriction in national and international travel, thus, stopping the spread of resistant bacteria, leading to a decrease in the AMR [54].

Limited data are available in the literature of comparative studies concerning resistance patterns of bacteria involved in the dynamics of UTIs. A recent study from Italy involved 83 positive urine and blood cultures from over 300 patients admitted to the hospital in two different periods of three months, before and after the COVID pandemic, respectively, similar to this study, has underlined that presence of MDR strains in post COVID patients is lower compared with patients before the pandemic [19].

Our study’s results determined fewer positive urine cultures in patients during pandemics compared with the same period, two years earlier. Yet, an overall decrease of patients’ presentation to the emergency room has also been observed; this matter could result from personal isolation and, from the medical point of view, loss of follow-up of chronic and uncomplicated cases. However, the same Italian paper suggested higher incidence in the overall MDR strains for most of the observed bacteria, which is similar to this study in terms of resistance rates to individual antimicrobials. The highest increased resistance was observed for *Pseudomonas* spp. and *Klebsiella* spp., two of the most treacherous microorganisms, with an increased capability of gaining resistance. Last year’s paper of Celeste Moya described the various mechanisms of adaptability of *Klebsiella pneumoniae* to multiple classes of antibiotics and concluded the concerning high number of pan-resistant strains of this Gram-negative bacterium [55]. Moreover, a review from 2019 described the adaptability of *Pseudomonas* spp. to most antibiotics and concluded that its eradication has become increasingly difficult due to its remarkable capacity to resist antimicrobials [56]. These data, corroborated with our findings, strengthen the idea of the possibility of creating “super-bugs” by continuing to overprescribe antibiotics on a daily basis, and to an even greater extent, when the indication is neither safe nor valuable for treating a viral infection such as COVID-19, generating high rates of resistance on short periods of time. However, we consider that part of the amplified rise in resistance, such as in *Pseudomonas* spp. for Carbapenems from R = 16% to over R = 60%, Ceftazidime from R = 27% to R = 67%, Levofloxacin from R = 30% to R = 77% and Klebsiella spp. to Levofloxacin from R = 16% to R = 35%, Ceftazidime from R = 16% to R = 25% or Amoxicillin-Clavulanic Ac. from R = 29% to 38% is an effect of the lowering access to medical treatment of chronic or non-urgent cases and prioritizing access to the hospital for more severely affected patients due to the pandemic, a fact that could also impact the outcomes of this study.

### 3.5. Limitations

The main limitation of the presented study is the limited number of processed urine cultures. The higher the estimate of the inspected probes is, the better outcomes the results will provide. Yet, this study depicts data from female patients with various urological and non-urological conditions from two highly rated university hospitals from the largest city of Romania. Therefore, extrapolating the results to the general population is feasible and most probably mirrors the reality of UTIs’ resistance rates in this demographic area.

Another limitation is represented by the impossibility of providing more data on patients’ medical history, such as the history of antibiotic consumption, previous surgeries on the urinary tract or record of indwelling catheters, key factors in developing and evolving resistant uropathogen strains.

A third limitation of this survey is that we cannot precisely tell if this trend would have been different in a non-pandemic situation, but, as other reports are still lacking, we find it important to disclose these results.

The short period between the two studied intervals and the limited time from the pandemic outbreak are also facts that could contribute to the bias of our conclusions. Further research is required for a better understanding of the dynamics of the viral disease in the emergence of antibiotics resistance in uropathogens. Regardless of the presented limitations, this study could represent pioneering research about the fundamental role that the pandemic is playing in selecting resistant strains of bacteria involved in UTIs, due to various previously discussed reasons, especially in the female patients at a higher risk of developing this condition.

## 4. Materials and Methods

### 4.1. Study Design and Sample Population

The study conducted was a descriptive “cross-sectional” retrospective study that engaged patients from two different university hospitals in Bucharest, Romania: “Prof. Dr. Th. Burghele” Clinical Hospital (BCH) and Elias University Hospital (EUH). The collected data implied an overall duration of 12 months, divided into two different periods of 6 months between 1 September 2018–28 February 2019, before the COVID19 pandemic and 1 September 2020 and 28 February 2021, after more than six months of the viral outbreak.

A total number of 24,155 urine probes were collected from the two centers enrolled in the study for bacterial analysis by processing standard urine culture, of which 2469 patients met the inclusion criteria. The representative diagram of patient dynamics is illustrated in Figure 1.

Data included in the study considered information such as sex and age for both hospitalized and non-hospitalized patients; thus, it was not possible to record a more extensive history for the latter.

### 4.2. Inclusion and Exclusion Criteria

The inclusion criteria

Positive uroculture ≥ 10^5^ CFU/mL;Single bacteria strain on the standard urine culture;Female patients;Age ≥ 18 years old.

The exclusion criteria:Less than 10^5^ CFU/mL on urine culture;Inoculation of more than one bacterial strain on urine culture;Male patients;Patients with urinary catheters

### 4.3. Sample Collection, Bacterial Culture, Identification of Uropathogens, Antibiotic Susceptibility Test

A judicious policy of antibiotic administration in the treatment of UTIs was applied in both studied centers, according to Romanian and European guidelines on urological infections [14,57]. For a proper urine culture analysis, probes were collected after at least 7–10 days of the antibiotic-free period.

In all cases, urine probes collection followed the international safety standards [58]. For the viable microorganism determination, inoculation and incubation were followed by bacterial identification that involved morphology, Gram-reactions and biochemical characteristics; detecting each bacterial strain’s sensitivity and resistance patterns on antibiogram followed the Clinical Laboratory Standard Institute (CLSI) [59] guidelines in most cases. Elias University Hospital (EUH) switched during last year to the European Committee of Antimicrobial Susceptibility Testing (EUCAST) [60] guidelines in terms of microbiology testing. Thus, probes collected during 1 September 2020 and 28 February 2021 in this hospital adhere to these guidelines; both practices are recommended in the World Health Organization’s Global Antimicrobial Resistance Surveillance System (GLASS) [61].

Bacterial culture, the identification of uropathogens, and the antibiotic susceptibility test used were previously described in more detail [9,11,15].

## 5. Conclusions

The presented study highlights *Escherichia coli* as the most ubiquitous bacteria involved in the epidemiology of UTIs in female patients, followed by *Klebsiella* spp.; *Enterococcus* spp. was the most frequent Gram-positive microorganism.

In most cases, the rates of resistance to various common antibiotics increased during the COVID pandemic, compared with the same period of the year, two years earlier. The increased resistance was observed for *Pseudomonas* spp. and *Klebsiella* spp. Levofloxacin, Amoxicillin-Clavulanic Ac. and Carbapenems showed the highest growth in resistance. The only examples for which decreasing resistance was observed are *Proteus* spp. for Aztreonam, *Enterococcus* spp. for Penicillin, and *Staphylococcus* spp. for Amikacin, Linezolid, and Nitrofurantoin.

## Figures and Tables

**Figure 1 antibiotics-11-00376-f001:**
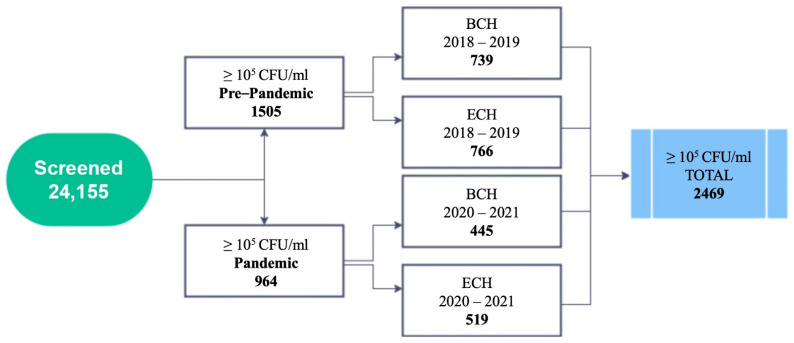
Diagram of the screened and enrolled patients in the study.

**Table 1 antibiotics-11-00376-t001:** Bacterial strains isolated in the study group.

Isolated Bacteria	BCH				EUH				Total				TOTAL	
	Pre-Pandemic		Pandemic		Pre-Pandemic		Pandemic		Pre-Pandemic		Pandemic			
	*n*	%	*n*	%	*n*	%	*n*	%	*n*	%	*n*	%	*n*	%
Gram-negative	584	79.02	368	82.69	697	90.99	483	93.06	1281	85.11	851	88.27	2132	86.35
*Escherichia coli*	400	54.12	234	52.58	525	68.53	346	66.66	925	61.46	580	60.16	1505	60.95
*Klebsiella* spp.	110	14.88	75	16.85	133	17.36	108	20.80	243	16.14	183	18.98	426	17.25
*Proteus* spp.	56	7.57	36	8.08	24	3.13	21	4.04	80	5.31	57	5.91	137	5.54
*Pseudomonas* spp.	18	2.43	23	5.16	15	1.95	8	1.54	33	2.19	31	3.21	64	2.59
Gram-positive	155	20.97	77	17.30	69	9.0	36	6.93	224	14.88	113	11.72	337	13.64
*Enterococcus* spp.	127	17.18	58	13.03	64	8.35	36	6.93	191	12.69	94	9.75	285	11.54
*Staphilococcus* spp.	28	3.78	19	4.26	5	0.65	-	-	33	2.19	19	1.97	52	2.1

*n*—number, %—percentage, BCH—Burghele Clinical Hospital, EUH—Elias University Hospital.

**Table 2 antibiotics-11-00376-t002:** Uropathogens in female patients of various age groups in BCH and EUH.

Isolated Bacteria	BCH												EUH											
	Pre-Pandemic						Pandemic						Pre-Pandemic						Pandemic					
	≤40		41–55		>55		≤40		41–55		>55		≤40		41–55		>55		≤40		41–55		>55	
	*n*	%	*n*	%	*n*	%	*n*	%	*n*	%	*n*	%	*n*	%	*n*	%	*n*	%	*n*	%	*n*	%	*n*	%
*E. coli*	66	8.93	60	8.11	274	37.07	38	8.53	55	12.35	141	31.68	50	6.52	44	5.74	431	56.26	39	7.51	46	8.86	261	50.28
*Klebsiella* spp.	15	2.02	16	2.16	79	10.69	14	3.14	14	3.14	47	10.56	9	1.17	9	1.17	115	15.01	7	1.34	13	2.50	88	16.95
*Proteus* spp.	11	1.48	9	1.21	36	4.87	9	2.02	9	2.02	18	4.04	-	-	2	0.26	22	2.87	1	0.19	2	0.38	18	3.46
*Pseudomonas* spp.	2	0.27	1	0.13	15	2.02	-	-	8	1.79	15	3.37	-	-	3	0.39	12	1.56	-	-	2	0.38	6	1.15
*Enterococcus* spp.	19	2.57	28	3.78	80	10.82	8	1.79	10	2.24	40	8.98	6	0.78	6	0.78	52	6.78	2	0.38	7	1.34	27	5.2
*Staphylococcus* spp.	4	0.54	8	1.08	16	2.16	2	0.44	6	1.34	11	2.47	2	0.26	-	-	3	0.39	-	-	-	-	-	-
Total	117	15.83	122	16.5	500	67.65	71	15.95	102	22.92	272	61.12	67	8.74	64	8.35	635	82.89	49	9.44	70	13.48	400	77.07

*n*—number, %—percentage, BCH—Burghele Clinical Hospital, EUH—Elias University Hospital.

**Table 3 antibiotics-11-00376-t003:** Antibiotic resistance profile in *Escherichia coli* and *Klebsiella* spp.

Antibiotics	*Escherichia coli*										*Klebsiella* spp.									
	Pre-Pandemic					Pandemic					Pre-Pandemic					Pandemic				
	R		S		NA	R		S		NA	R		S		NA	R		S		NA
	*n*	%	*n*	%	%	*n*	%	*n*	%	%	*n*	%	*n*	%	%	*n*	%	*n*	%	%
Amikacin	34	3.67	809	87.45	8.86	28	4.82	470	81.03	14.13	26	10.69	217	89.3	-	35	19.12	145	79.23	1.63
Amoxicillin– Clavulanic Ac.	132	14.27	718	77.62	8.1	124	21.37	373	64.31	14.31	71	29.21	169	69.54	1.23	71	38.79	105	57.37	3.82
Aztreonam	30	3.24	362	39.13	57.67	23	3.96	170	29.31	66.72	24	9.87	82	33.74	56.37	24	13.11	45	24.59	62.29
Ceftazidime	66	7.13	766	82.81	10.05	47	8.1	449	77.41	14.48	39	16.04	198	81.48	2.46	46	25.13	136	74.31	0.54
Fosfomycin	2	0.21	847	91.56	8.21	4	0.68	533	91.89	7.41	3	1.23	46	18.93	79.83	-	-	-	-	-
Imipenem	0		404	43.67	56.32	3	0.51	262	45.17	54.31	11	4.52	205	84.36	11.11	14	7.65	134	73.22	19.12
Levofloxacin	254	27.45	587	63.45	9.08	167	28.79	329	56.72	14.48	41	16.87	199	81.89	1.23	65	35.51	115	62.84	1.63
Meropenem	1	0.1	414	44.75	55.13	1	0.17	267	46.03	53.79	18	7.4	203	83.53	9.05	21	11.47	128	69.94	18.57
Nitrofurantoin	45	4.86	631	68.21	26.91	39	6.72	411	70.86	22.41	38	15.63	105	43.2	41.15	38	20.76	72	39.34	39.89

R—resistant, S—sensitive, NA—not available, *n*—number, %—percentage.

**Table 4 antibiotics-11-00376-t004:** Antibiotic resistance profile in *Pseudomonas* spp. and *Proteus* spp.

Antibiotics	*Pseudomonas* spp.										*Proteus* spp.									
	Pre-Pandemic					Pandemic					Pre-Pandemic					Pandemic				
	R		S		NA	R		S		NA	R		S		NA	R		S		NA
	*n*	%	*n*	%	%	*n*	%	*n*	%	%	*n*	%	*n*	%	%	*n*	%	*n*	%	%
Amikacin	5	15.15	27	81.81	3.03	16	51.61	14	45.16	3.22	7	8.75	73	91.25	-	7	12.28	48	84.21	3.5
Amoxicillin– Clavulanic Ac.	6	18.18	10	30.3	51.51	8	25.8	2	6.45	67.74	22	27.5	55	68.75	3.75	16	28.07	33	57.89	14.03
Aztreonam	3	9.09	15	45.45	45.45	13	41.93	7	22.58	35.48	4	5.0	51	63.75	31.25	2	3.5	29	50.87	45.61
Ceftazidime	9	27.27	23	69.69	3.03	21	67.74	10	32.25	-	12	15.0	68	85.0	-	9	15.78	48	84.21	-
Fosfomycin	6	18.18	26	78.78	3.03	21	67.74	10	32.25	-	0	0	55	68.75	31.25	4	7.01	32	56.14	36.84
Imipenem	10	30.3	22	66.66	3.03	24	77.41	6	19.35	3.22	19	23.75	55	68.75	7.5	19	33.33	33	57.89	8.77
Levofloxacin	6	18.18	26	78.78	3.03	20	64.51	10	32.25	3.22	1	1.25	57	71.25	27.5	1	1.75	42	73.68	24.56
Meropenem	0	0	10	30.3	69.69	3	9.67	4	12.9	77.41	-	-	-	-	-	-	-	-	-	-
Nitrofurantoin	5	15.15	27	81.81	3.03	16	51.61	14	45.16	3.22	7	8.75	73	91.25	-	7	12.28	48	84.21	3.5

R—resistant, S—sensitive, NA—not available, ***n***—number, %—percentage.

**Table 5 antibiotics-11-00376-t005:** Antibiotic resistance profile in *Enterococcus* spp. and *Staphylococcus* spp.

Antibiotics	*Enterococcus* spp.										*Staphylococcus* spp.									
	Pre-Pandemic					Pandemic					Pre-Pandemic					Pandemic				
	R		S		NA	R		S		NA	R		S		NA	R		S		NA
	*n*	*%*	*n*	*%*	*%*	*n*	*%*	*n*	*%*	*%*	*n*	*%*	*n*	*%*	*%*	*n*	*%*	*n*	*%*	*%*
Amikacin	-	-	-	-	-	-	-	-	-	-	1	3.03	31	93.93	3.03	0	0	6	31.57	68.42
Ampicillin	34	17.8	147	76.96	5.23	17	18.08	75	79.78	2.12	-	-	-	-	-	-	-	-	-	-
Trimetoprim/Sulfamethoxazol	-	-	-	-	-	-	-	-	-	-	6	18.18	19	57.57	24.24	1	5.26	17	89.47	5.26
Ceftazidime	-	-	-	-	-	-	-	-	-	-	7	21.21	20	60.6	18.18	-	-	-	-	-
Fosfomycin	1	0.52	135	70.68	28.79	6	6.38	56	59.57	34.04	-	-	-	-	-	-	-	-	-	-
Levofloxacin	61	31.93	123	64.39	3.66	33	35.1	57	60.63	4.25	7	21.21	22	66.66	12.12	5	26.31	12	63.15	10.52
Linezolid	4	2.09	165	86.38	11.51	4	4.25	77	81.92	13.82	2	6.06	25	75.75	18.18	0	0	17	89.47	10.52
Nitrofurantoin	6	3.14	166	86.91	9.94	4	4.25	86	91.48	4.25	2	6.06	24	72.72	21.21	1	5.26	15	78.94	15.78
Penicillin	56	29.31	111	58.11	12.56	24	25.53	58	61.7	12.76	15	45.45	11	33.33	21.21	9	47.36	9	47.36	5.26
Vancomycin	3	1.57	171	89.52	8.9	2	2.12	79	84.04	13.82	-	-	-	-	-	-	-	-	-	-

R—resistant, S—sensitive, NA—not available, *n*—number, %—percentage.

## Data Availability

Data supporting the reported results are available from the authors.

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
