# Peer review of "Does the COVID Pandemic Modify the Antibiotic Resistance of Uropathogens in Female Patients? A New Storm?"

_antibiotics, 2022, doi:10.3390/antibiotics11030376_

Round 1
Reviewer 1 Report
This article demonstrated the change of distribution of uropathogens and antibiotic resistance between pre-pandemic and pandemic era. It would provide us important information regarding contemporary treatment for UTI. However, there are some issues to be addressed.
Major
- The order of sections should revised, i.e. 1. introduction, 2. materials and methods, 3. results, 4. discussion, and 5. conclusions.
- How many patients with positive urine culture for multiple bacteria were excluded from this study? If large number of patients were excluded, current results might not reflect the fact. The authors should show the number of those patients.
Minor
- Were similar results shown in the trend of sputum or blood culture during the pandemic era? If so, the current results would be more strengthened.
Reviewer 2 Report
In this study, you have shown that the bacterial resistanceto antibiotics increased significantly during the pandemic.
Are data available to show that this increase
was faster than in previous observed periods ?
Otherwise how can be supported the concept that this was an increase surely
induced by the pandemic and that it would be happen anyway ?
Why did the study focused the attention on the resistance of uropathogens to the antibiotics
and not of bacteria affecting the respiratory system ? Is this because the UTI
can be considered a sort of "model" to assess the increase of bacterial
resistance to antibiotics ?
I think you should explain to the readers these concepts because not all are
capable to deeply understand the concepts.
1) whether the increase in resistance to antibiotics by bacteria
in the two years of the pandemic was greater than the increase that
was observed in previous periods studied
and that would have been observed without COVID
2) If the resistance data observed for urinary infections can be used /
extrapolated generally to all infections of the other systems and then
the conclusions can be generalized.
I think that part of the discussion could also be dedicated
to suggest what epidemiological, clinical and therapeutic strategies
should be adopted to limit the development of bacterial resistance
to antibiotics. In addition, some comments could be added about the research
for new drugs to overcome these resistances, and the current therapeutical choices.
At row 184 I think the sentence is incomplete....
At row 256 a "s" is lacking in the word Klebiella
Round 2
Reviewer 1 Report
The manuscript has been appropriately revised by answering reviewers' comments.